# Probiotic Adhesion to Skin Keratinocytes and Underlying Mechanisms

**DOI:** 10.3390/biology11091372

**Published:** 2022-09-19

**Authors:** Mariana Lizardo, Rui Miguel Magalhães, Freni Kekhasharú Tavaria

**Affiliations:** Centro de Biotecnologia e Química Fina–Laboratório Associado, Escola Superior de Biotecnologia, Universidade Católica Portuguesa/Porto, Rua Diogo Botelho, 1327, 4169-005 Porto, Portugal

**Keywords:** probiotics, keratinocytes, skin, adhesion, HaCaT cells

## Abstract

**Simple Summary:**

The use of probiotics to ameliorate skin conditions has been suggested. This is based in the fact that they compete with pathogenic bacteria for adhesion sites, thereby displacing unwanted microorganisms. *Lacticaseibacillus rhamnosus* was able to adhere effectively to keratinocytes decreasing the number of adherent pathogenic bacteria. In the presence of pathogens all tested probiotics decreased invasion by *S. aureus*, one of the most relevant skin pathogens. *Ex vivo* models also showed wound healing capacity of *L. rhamnosus* with a concomitant decrease in the viable numbers of *S. aureus*, suggesting it is a good candidate as a co-adjuvant in the treatment of skin infections by this pathogen.

**Abstract:**

The effects of probiotics on the skin are not yet well understood. Their topical application and benefits derived thereafter have recently been investigated. Improvements in different skin disorders such as atopic dermatitis, acne, eczema, and psoriasis after their use have, however, been reported. One of the mechanisms through which such benefits are documented is by inhibiting colonization by skin pathogens. Bacterial adhesion is the first step for colonization to occur; therefore, to avoid pathogenic colonization, inhibiting adhesion is crucial. In this study, invasion and adhesion studies have been carried out using keratinocytes. These showed that *Escherichia coli* is not able to invade skin keratinocytes, but adhered to them. *Lacticaseibacillus rhamnosus* and *Propioniferax innocua* decreased the viable counts of the three pathogens under study. *L*. *rhamnosus* significantly inhibited *S*. *aureus* adhesion. *P. innocua* did not inhibit pathogenic bacteria adhesion, but when added simultaneously with *S. aureus* (competition assay) a significant adhesion reduction (1.12 ± 0.14 log_10_CFU/mL) was observed. Probiotic bacteria seem to use carbohydrates to adhere to the keratinocytes, while *S*. *aureus* uses proteins. *Lacticaseibacillus rhamnosus* showed promising results in pathogen inhibition in both in vitro and ex vivo experiments and can potentially be used as a reinforcement of conventional therapies for skin dysbiosis.

## 1. Introduction

The skin is the biggest organ in the human body, playing a crucial role in protecting against external damage, regulating temperature, as well as water and salt loss [1]. It is a complex ecosystem that supports the growth of indigenous microbiota [2] consisting of bacteria, fungi, viruses, and mites, coexisting with the host and contributing to tissue integrity and immune homeostasis. The majority of skin bacteria are harmless and provide a diverse microbiome that helps in the resolution of disease. However, on the other side, the perturbation of this microbiota can influence normal skin health and predisposes to pathogenic colonization leading to inflammatory dermatological disorders [3].

The epidermis is mostly composed of one single cell type—the keratinocyte, in different stages of differentiation. These cells comprise over 95% of the cell mass of the outermost portion of the skin and were first studied for their contribution to the structural integrity (through keratin production) and barrier formation of the skin through lipid biosynthesis [4]. The immortalized human keratinocyte line HaCaT (human, adult, low calcium, high temperature) is used as a substitute for normal human keratinocytes in vitro due to its highly preserved differentiation capacity, being a useful tool in human cell research. HaCaT cells demonstrate basal cell properties and still respond to various inducers of differentiation such as Ca^2+^ and high cell density, can also form a nearly normal epithelial structure when transplanted, can be grown in organotypic cultures, and can be transfected at moderate levels [5]. The HaCaT cell line was used in this study because it closely resembles primary keratinocyte cells, having been used before in studies of different bacteria and keratinocyte interactions [6].

Pathogenic bacteria like *Pseudomonas aeruginosa*, *Staphylococcus aureus*, and *Escherichia coli* can colonize the skin and soft tissues, disrupting the microbiome and leading to unwanted conditions [7]. *Pseudomonas aeruginosa* is an opportunistic pathogen, being the major cause of infection in immunocompromising patients, with cancer and patients suffering from severe burns or cystic fibrosis [8]. Responsible for superficial infections such as folliculitis or pyodermitis, external otitis, or ulcerous keratitis [9], *Staphylococcus aureus* is a nonmotile and Gram-positive cocci. Humans are a natural reservoir of this pathogen with a niche preference for the anterior nares. It is involved in skin infections of atopic dermatitis patients, being present in more than 90% of the inflammatory lesions colonized by this pathogen [10]. *Escherichia coli* is a Gram-negative rod shape bacterium that typically colonizes the gastrointestinal tract of humans and warm-blooded animals [11]. According to various reports, *E. coli* can be involved in skin and soft tissue infections (SSTIs), being frequently isolated from these in patients with underlying diseases, particularly in immunocompromised patients [12]. In this work, we address the issue of pathogen colonization of keratinocytes by using probiotics as potential candidates to compete for adhesion sites. After in vitro validation, the experiment was validated using human skin equivalent models to mimic conditions that were more approximated to reality.

Tissue-engineered models that simulate human skin, also known as human skin equivalents (HSEs), are three-dimensional in vitro tissues that mimic the natural tissue and are great research tools to mimic real skin conditions [13]. HSEs can partially replicate physiological skin functions, such as proliferative capacity, extracellular matrix synthesis, cellular signaling, and responses to numerous stimuli. They have been promoted as a promising alternative to animal experiments for safety assessment in skin and drug research and are also very useful as clinical skin replacements and grafts [14]. HSEs are constituted by human skin cells (keratinocytes, fibroblasts, and stem cells) and elements of the extracellular matrix (mostly collagen), and can be designed as epidermis only, dermis only, or full thickness [15].

Although probiotics have been widely studied to treat and prevent gastrointestinal disorders, some gastrointestinal diseases can show changes at the skin level, evidencing a possible link between gut dysbiosis and dermatological diseases [16] through the gut–brain–skin axis theory. The skin manifestations may even precede clinically evident gastrointestinal disease [17]. Some authors [18] suggest that microbiome modulation through the administration of probiotics could have beneficial effects on skin homeostasis and skin inflammation, among others, later reporting that even if the main target has been the gastrointestinal tract, oral probiotics can affect other conditions besides the gut. Later, clinical trials suggested that probiotics do not exert their beneficial effects only by the gastrointestinal route but also through topical applications [18].

However, before such solutions can be applied topically, in vitro studies such as this one are needed to evaluate invasion and adhesion onto keratinocytes. A human skin model was then used to mimic the process closer to a real skin-like situation.

## 2. Materials and Methods

### 2.1. Bacterial Strains and Culture Conditions

The probiotic strains used in this study were *Lactobacillus delbrueckii* subsp. *bulgaricus* DSM 20081, *Lacticaseibacillus rhamnosus* DSM 20021, and *Propioniferax innocua* DSM 8251 and the pathogenic strains were *Escherichia coli* DSM 1103, *Pseudomonas aeruginosa* DSM 1117, and *Staphylococcus aureus* DSM 799, all from DSMZ (Deutsche Sammlung von Mikroorganismen und Zellkulturen, Braunschweig, Germany). The strains were obtained as frozen stocks at −78 °C, then cultured on de Man, Rogosa, and Sharp broth (MRS, Biokar diagnostics, Allone, France) media, for lactic acid bacteria, or brain-heart infusion broth (BHI, Biokar diagnostics), for pathogenic bacteria, and incubated at 37 °C for 24 h.

### 2.2. Cell Culture Assays (HaCaT)

The immortalized human keratinocyte line HaCaT obtained from Cell lines Service (Oppenheim, Denmark) was defrosted from liquid nitrogen and maintained in a composed Dulbecco’s Modified Eagle’s Medium (DMEM) high glucose with L-glutamine (Gibco, Waltham, MA, USA) plus 10% (*v*/*v*) of fetal bovine serum (FBS, BioWest, Nuaillé, France) and 1% (*v*/*v*) of penicillin-streptomycin-fungizone solution (pen-strep, Lonza, Veriers, Belgium). Cells were incubated as monolayers first in 25 cm^2^ cell culture flasks (Sarstedt, Nümbrecht, Germany) after defrosting, scaled up in 75 cm^2^ flasks, and stored in a humidified atmosphere of 95% air and 5% of CO_2_.

The medium was substituted with 2 d intervals, and whenever 80% of cells’ confluence was reached, they were washed three times with sterile phosphate-buffered saline (PBS) approximately 90 µL/cm^2^ of the vessel surface, and then detached with TryPLE^TM^ Express (Gibco, Waltham, MA, USA) 40 µL/cm^2^ and incubated for 15 min. Subsequently, a complete DMEM medium was added, and a repeated up and down movement with the aid of a pipette was made until all the cells detached. Once the detachment was reached, the cells were transferred to a falcon tube and centrifugated for 10 min at 120× *g* to sediment; the supernatant was discarded and the cells were resuspended in 1 mL of DMEM complete medium, to proceed to viable cell counting using trypan blue (Lonza, Verviers, Belgium) in a TC20^TM^ Automated Cell Counter (Bio-Rad, Hercules, CA, USA). Then the cells were seeded in new T-75 flasks or in a 24-well microplate and later used in assays after 48 h of incubation.

During the cell culture assays, the HaCaT cells were used between passages 37 and 72. In these assays, the cells had a density of 2.0 × 10^5^ cells/mL (measured by the trypan blue assay), and each experiment was repeated three times, and each bacterium or combination was performed in triplicate.

### 2.3. Invasion Assays

Before the beginning of the assay, the complete cell culture medium was changed to only DMEM medium, without the addition of FBS and pen-strep, and the cells were incubated for a minimum of 30 min.

To find out if the probiotics (*L*. *rhamnosus*, *L*. *delbrueckii,* and *P*. *innocua*) inhibited keratinocytes invasion by the pathogenic bacteria (*E*. *coli*, *S*. *aureus,* and *P*. *aeruginosa*), the cells were exposed for 1 h to each bacterium strain alone and with the combinations of probiotics plus the pathogenic bacteria [19]. All bacterial solutions were plated in the correspondent culture medium (MRS for probiotic, BHI for pathogenic, and both for combinations). After incubation, the cells were washed three times with PBS, to remove the non-adherent bacteria, and the medium was modified for DMEM plus 20 µg/mL of gentamicin sulfate (Lonza^TM^ BioWhittaker^TM^, Visp, Switzerland) and incubated for 1 h 30 min. Next, the cells were washed three times with PBS and lysed in a solution with 0.2% of Triton (Sigma-Aldrich, St. Louis, MO, USA). After vigorous and persistent up and down, the content of each well was collected into Eppendorf tubes that were rapidly mixed in a vortex, serially diluted, and counted.

### 2.4. Adhesion Assays

As previously said, prior to the assay the keratinocyte medium was changed to a medium without antibiotic, and the bacterial concentration was adjusted to 10^6^ CFU/mL. since this bacterial concentration was utilized in similar studies [19]. To discover what was the effect of the probiotic bacteria on the keratinocyte adhesion by pathogens, the protocol was adapted from [19], and some different experiments were performed; exclusion where the cells were exposed to the probiotic strain (*L*. *rhamnosus*, *L*. *delbrueckii* or *P*. *innocua*) and after one hour of incubation the pathogenic strain was added (*E*. *coli*, *S*. *aureus* and *P*. *aeruginosa*); substitution where cells were first exposed to the pathogen and 30 min later to the probiotic; and competition in which the bacterial strains were added at the same time to the cells. After incubation, the cells were washed three times with PBS and TryPLE^TM^ Express was added for cell detachment, then each well content was collected, and adherent bacteria were counted using serial dilutions.

### 2.5. Carbohydrate Analysis

To determine the role of carbohydrates in probiotic adhesion to keratinocytes, the bacteria were exposed to a solution of 50 mM of sodium-metaperiodate (Sigma-Aldrich, St. Louis, MO, USA) in 0.1 M citrate phosphate buffer (pH 4.5), and as the control, the bacteria were treated only with the buffer. Then the bacteria were placed in a Thermoblock at 37 °C for 30 min, and after this time washed with PBS and prepared to infect the cells and a regular adhesion assay was carried out.

### 2.6. Protein Analysis

To find out if the adhesion to HaCaT cells by *S*. *aureus* and *L*. *rhamnosus* was carried out by an adhesin protein, the bacteria were centrifuged and resuspended in proteinase K solution 20 mg/mL (Frilabo, Maia, Portugal) or 0.4% trypsin (American Type Culture Collection, Manassas, VA, USA). The control was prepared with PBS only, and the bacteria were placed at 37 °C for 2 h. After this time, the bacteria were washed with PBS and placed in cell culture medium. The cells were exposed to the bacteria and incubated for 1 h, after which they were washed three times with PBS and detached with TryPLE^TM^ Express. The counting of adherent bacteria was performed after serial dilutions.

### 2.7. Human Skin Equivalent Assays

With the aim of understanding the effect of the probiotic *L*. *rhamnosus* in skin infected by the pathogen *S*. *aureus*, four human skin equivalents (Phenion^®^ Large FT skin model) were obtained from an expert company (Phenion, Henkel, Düsseldorf, Germany). To obtain epidermal wounding, the models were scratched with an 8 mm dermal biopsy punch (Razormed^TM^, Gurugram, India) and afterwards, prepared according to the manufacturer’s instructions and 8 mL of air–liquid interface medium (ALI, Phenion^®^) were added.

For wound infection, the bacteria were inoculated (25 µL) with approximately 10^6^ CFU/mL of both bacteria, with the following organization: one of the models only with *S*. *aureus*, the other with *L*. *rhamnosus*, one with the pathogenic and probiotic addition with 24 h delay, and a control with no bacterial addition. Each of the models infected with the pathogen had two wounds each, and the remaining only one.

During the 11 days of the experiment, the skin models were incubated at 37 °C with 5% CO_2_, and the ALI medium was changed every 24 h if necessary, or every 48 h, as needed. Pictures of the wound evolution were taken every day at approximately the same time. For assessment of the wound infection, some samples were collected from the infected wounds utilizing sterile swabs and homogenized in PBS. The bacterial load of the skin model wounds was evaluated by serial dilutions in MRS (*L*. *rhamnosus*) and BHI (*S*. *aureus*) media. On the 11th day, the skin models were sacrificed and put on a stomacher for 2 min, to rupture the skin models. Then serial dilutions were made, and the bacteria were counted.

### 2.8. Statistical Analysis

The data analysis for the invasion, adhesion, and mechanisms of adhesion assays was done using IBM SPSS (IBM, New York, NY, USA), using one-way ANOVA, and when necessary, multiple comparisons using the post hoc Tukey test. The significance level considered was 0.05.

## 3. Results and Discussion

### 3.1. Invasion Assays

The capacity of the three pathogenic bacteria under study to internalize into HaCat cells was explored. Invasion plays an important role especially in maintaining persistent or recurring infection, although in order to invade eukaryotic cells the pathogenic bacteria have first to attach to the cell surface [6].

As can be seen in Figure 1, *E. coli* was not capable of invading HaCaT cells efficiently. This result can be justified by the fact that *E*. *coli* is not well recognized as a skin pathogen since in healthy human skin colonization by this bacterium is rare [20]. According to the authors, keratinocytes secrete the S100 protein psorian, an *E*. *coli*-killing antimicrobial protein that belongs to a calcium-responsive signaling proteins group [21].

*Pseudomonas aeruginosa* could successfully invade the HaCaT culture. This bacterium is known to be a versatile opportunistic pathogen present in wound infections, occurring mostly after injuries such as burns or chronic cutaneous wounds [22]. According to the literature, *P*. *aeruginosa* is characterized by high bacterial invasion of keratinocytes, decreasing the cell viability [23]. *Pseudomonas aeruginosa* is also described as being cytotoxic inhibiting cell migration in HaCaT cell culture [24]. In a model of reconstructed human epidermis, *P*. *aeruginosa* was found in the deepest epidermis layer (*stratum basale*), proving the high skin invasion of this pathogen [25].

*Staphylococcus aureus* displayed the highest invasion capability of HaCaT cells, although close to that shown by *P*. *aeruginosa* presenting no significant differences (*p* > 0.05). Being responsible for a wide range of superficial and invasive infections that can be mild to fatal, *S*. *aureus* is a bacterium that can persist within keratinocytes for long periods of time [6]. Keratinocyte invasion by *S. aureus* has been studied by various authors, revealing a high value of invading bacteria [26]. Edwards et al. (2011) also proved that *S. aureus* presented significant invasion differences between different cell lines such as HaCaT and endothelial cells, obtaining lower values for keratinocyte invasion. The low values for pathogenic invasion may be justified by the short time of cell exposition to the bacteria, only 1 h; increasing this time perhaps the number of invading bacteria would be higher.

Figure 2 shows the capacity of probiotic bacteria to invade the HaCaT cells; from the results, it is evident that the probiotic bacteria, particularly the *Lactobacillus* and *Lactocaseibacillus* genera, obtained higher HaCaT invasion values than the pathogenic bacteria. *P*. *innocua* had a significantly lower invasion value compared to the other probiotic bacteria (*p* < 0.05 in both cases). However, these results cannot be compared with those found in the literature, because the existing studies with probiotics and keratinocyte invasion do not focus on the probiotic invasion itself, but on the pathogen’s behavior influenced by probiotic bacteria; these studies evaluated the probiotic capacity to inhibit the internalization of the pathogen through bacterial counting of the pathogenic bacteria versus coinfection with the pathogen and probiotic [27,28].

In co-cultures (Figure 3), *E*. *coli* showed no ability to invade keratinocytes in the presence of probiotic bacteria; but this result could be related to the inoculum concentration. The initial counts were higher and the obtained values for *E*. *coli* invasion were not detectable (below the method’s detection limit). However, also, the possible probiotic inhibition effects on *E. coli* must be considered, since both *Lactobacillus* strains used in this study, especially *L. rhamnosus*, had been reported to prevent *E*. *coli* infection in the gastrointestinal tract [29].

*Pseudomonas aeruginosa* invasion in the presence of *L*. *rhamnosus* resulted in a lower number of cells internalized compared with the remaining probiotic bacteria; however, this difference was not statistically significant (*p* > 0.05). The antimicrobial activity of this probiotic against *P*. *aeruginosa* has been studied previously, revealing positive results in a study where the pathogen was isolated from burn and wound infections, and analyzed with the plate diffusion method [30].

*Propioniferax innocua* did not invade keratinocytes in the presence of *P*. *aeruginosa*. This inhibition by *P*. *aeruginosa* could be related to some bacteriocin production or other defense mechanism of the pathogen, being only a hypothesis since the cell invasion interaction between these two bacteria is not well documented. However, in the study performed by Lopes et al. [31], it was also reported that *P*. *innocua* destroyed *P*. *aeruginosa* pre-formed biofilm. Nevertheless, lactobacilli revealed no significant differences in keratinocyte invasion in the presence of the pathogen (*p* > 0.05).

Analyzing the results for *S. aureus* HaCaT invasions (Figure 4), the lower invasion value was obtained with *L*. *delbrueckii,* while the highest invasion value was obtained for *L. rhamnosus. Lactobacillus rhamnosus* is susceptible to gentamicin [32], showing that antibiotic resistance is not the cause for higher invasion. Hor and Liong [33] reported that the cell-free supernatant of *L*. *delbrueckii* considerably impeded the biofilm formation of *S. aureus*, remembering that LAB are able to produce antimicrobial metabolites comprising lactic acid, acetic acid, hydrogen peroxide, and diacetyl that may inhibit pathogenic bacteria. Mohammedsaeed et al. [28] using a different cell line, normal human epidermal keratinocytes (NHEK), discovered that in the presence of the pathogen and probiotic, the viability of keratinocytes increased when compared to keratinocytes exposed only to the pathogen. Overall, invasion by the probiotic was always higher than for the pathogen invasion, demonstrating that the three probiotic strains in the study may be effective in reducing invasion by *S. aureus* when administered at the same time.

### 3.2. Adhesion Assays

Bacterial adhesion is a crucial first step in colonization, becoming a mandatory study target in bacterial pathogenesis. To effectively establish infection, the bacterial pathogens require adhesion to host cells, colonization of the tissues, and cellular invasion (in some cases), followed by intracellular proliferation, dissemination to other tissues, and persistence [34].

Bacterial pathogens express several molecules and structures capable of promoting cell attachment, denominated adhesins that act as a bridge between the bacteria and the host. So, to evaluate if the probiotic strains (*Lacticaseibacillus rhamnosus*, *Lactobacillus delbrueckii* subsp. *bulgaricus,* and *Propioniferax innocua*) can inhibit the HaCaT attachment by the pathogens under study (*Escherichia coli*, *Pseudomonas aeruginosa,* and *Staphylococcus aureus*), adhesion assays were performed using the three proposed models: exclusion, competition, and displacement. For exclusion, the cells were inoculated first with the probiotic and then with the pathogen; competition by adding the bacteria at the same time; and displacement by first putting the cells in contact with the pathogen and then with the probiotic [19].

Figure 5 shows *E. coli* adhesion in the presence of the three probiotics. *E*. *coli* adhered to HaCaT cells, either alone or in the presence of probiotics, although as said previously this pathogen proved not to invade keratinocytes. The obtained results for adhesion are in concordance with the results obtained by Lopes et al. [31] that used human keratin and reported cell adhesion by *E*. *coli.*

None of the probiotics inhibited pathogen attachment as a pre-treatment, since in the exclusion assay higher numbers of attached *E*. *coli* were observed, with *L*. *rhamnosus* and *L*. *delbrueckii* increasing by 0.62 and 0.48 log CFU/mL the pathogen attachment, respectively, although showing no statistically significant differences in relation to the control group (*p* > 0.05). However, on the other side, the probiotics proved effective in competition assays, specifically *P*. *innocua* and *L*. *rhamnosus* since a clear decrease in the viable counts of pathogenic bacteria was registered, from 4.64 ± 0.13 to 2.87 ± 0.26 and 2.68 ± 0.17 log CFU/mL, respectively. These differences were significant (*p* < 0.05 in both cases). *L*. *delbrueckii* showed the greatest effect in displacement, the unique situation for this bacterium with significant differences compared to the control (*p* < 0.05), indicating that this probiotic is effective if added after the pathogen; however, the obtained value (3.57 ± 0.74 log CFU/mL) is very close to that achieved for the other probiotic bacteria, 3.8 ± 0.51 and 3.75 ± 0.1 log CFU/mL for *L*. *rhamnosus* and *P*. *innocua*, respectively, making the difference within the tested probiotic bacteria, non-significant (*p* > 0.05). This suggests that all the used probiotic strains could potentially be helpful if used as a topical treatment after pathogen adhesion.

Regarding the probiotic behavior in the adhesion to HaCaT cells in the presence of *E*. *coli, L*. *rhamnosus* is the probiotic that mostly adhered to the HaCaT cells but is also more affected by the pathogen’s presence. A substantial decrease in exclusion and displacement assays (1.41 and 1.86 log CFU/mL, respectively) was apparent when compared to the control, where only the attachment of the probiotic to keratinocytes was tested, and also verified through statistical analysis to be significant (*p* < 0.05). This probiotic also decreased the adhesion in the competition assay (0.73 log CFU/mL) although with no statistical significance (*p* > 0.05). *P. innocua* revealed a considerable adhesion decrease in the displacement assay of 0.88 log CFU/mL in relation to the control (*p* < 0.01), while *L*. *delbrueckii* showed no significant differences under any of the studied conditions.

*Pseudomonas aeruginosa* adhesion inhibition (Figure 6) was not potentiated by the presence of any of the probiotic bacteria under study either used as pre-treatment or post-treatment, revealing no substantial differences between the control group and the revealed influence of the probiotic strains in the pathogen adhesion (*p* > 0.05).

However, in competition assays, *L*. *rhamnosus* and *P*. *innocua* were able to significantly (*p* < 0.05) decrease pathogen adhesion by 1.39 and 1.41 log CFU/mL, respectively, proving that if added at the same time, these probiotics have positive and promising effects. *L*. *delbrueckii* did not provoke any reduction in *P*. *aeruginosa* attachment in any of the experiments, indicating that this probiotic cannot be used as a treatment for *P*. *aeruginosa* infection. With respect to probiotic attachment in the presence of *P*. *aeruginosa*, *L*. *rhamnosus* lost largely the capacity to adhere to HaCaT cells when the pathogen was present, showing significant differences relative to control in all the experiments (*p* < 0.05). *Propioniferax innocua* also showed a high decrease in relation to the control in the exclusion assay (*p* < 0.01), while *L*. *delbrueckii* did not present any significant difference in adhesion in any experiment with the presence of the pathogen compared to the control group (*p* > 0.05).

Analyzing the obtained results for *S*. *aureus* (Figure 7), it can be concluded that is the pathogen under study that mostly adhered to HaCaT cells. This fact is in concordance with the study performed by Lopes et al. [31], where the three pathogenic strains used were identical, and *S*. *aureus* proved with a large difference to adhere more to human keratin. According to the authors, this fact could be related to the recognized ability of *S. aureus* to adhere to the human extracellular matrix and serum components due to the existence of adhesins [31].

With respect to the inhibition of pathogenic attachment in the presence of probiotics, *L*. *rhamnosus* demonstrated a higher influence in *S*. *aureus* adhesion in the displacement assay, with significant reduction compared to the control (*p* < 0.01) indicating that this probiotic is promising as a post-treatment. *Lactobacillus delbrueckii* also revealed the best results in displacement assays although with less attachment inhibition by *S*. *aureus* than *L*. *rhamnosus*, and with no significant differences in relation to control (*p* > 0.05). Contrary, *P. innocua* showed an equal pathogenic adhesion decrease in exclusion and displacement assays, although an extremely low effect, as can be seen, and without statistical relevance (*p* > 0.05). A more accentuate and significant (*p* < 0.05) *S*. *aureus* adhesion reduction (1.12 log CFU/mL) in competition assays was observed.

*Lacticaseibacillus rhamnosus* decreased the cell attachment significantly in the assays with pathogens in an identical way for the three assays (*p* < 0.05, in all the experiments), while *L*. *delbrueckii* did not significantly reduce the adhesion to HaCaT cells in the presence of the pathogens (*p* > 0.05). *Propioniferax innocua* did not decrease the attachment in exclusion assays but on the remaining it decreased considerably (for displacement 1.36 log CFU/mL, and competition 1.19 log CFU/mL).

### 3.3. Adhesion Mechanisms

The possible mechanisms behind bacterial adhesion were investigated. Different probiotic mechanisms of adhesion to keratin, the main protein of the skin, have been proposed as nonspecific interactions, and hydrophobic interactions [31]. In this study, carbohydrate and protein analysis were performed, which remains to be the principal main driving forces behind the strengthening of bacteria interfaces and hosts.

### 3.4. Carbohydrate Analysis

In order to investigate the possibility of a cell-surface carbohydrate being implicated in the mechanism of the studied probiotic strains’ adhesion to HaCaT cells, an assay utilizing sodium-meta periodate was performed. This is a chemical compound that oxidizes cell surface carbohydrates, being considered a useful tool to explore the structures of carbohydrates and compounds containing oxidizable functions. It has been used in studies to investigate probiotic adhesion to cells [19]. The procedure was made as indicated previously and the obtained results for this assay are present in Figure 8.

Sodium-metaperiodate significantly reduced the adhesion of the three probiotic strains to HaCaT cells, testified by statistical analysis (*p* < 0.01) for the three probiotics with treatment versus probiotics with buffer (control). Utilizing the plate count method only, the viable adherent bacteria were able to be counted. A lower number of adherent bacteria in the pre-treated group compared to the control is indicative that this compound may inhibit (through oxidation) the carbohydrates that could be behind the probiotic adhesion to keratinocyte cells. Nevertheless, according to Prince [19], this chemical compound affected bacterial viability. Oxidation of carbohydrates by meta periodate seems to have other effects besides elimination of cell surface polysaccharides. The author, set as an example the peptidoglycan structural molecule that could be oxidized, disrupting the cell wall and destroying the bacteria. There is evidence in this study to support that periodate-treated cells adhered less, meaning that the probiotic adhesion to keratinocytes was dependent on cell surface carbohydrates.

### 3.5. Protein Analysis

Next, the possibility that *L. rhamnosus* and *S. aureus* possessed cell surface proteins implicated in bacterial adhesion to HaCaT cells was investigated. For that, the protein experiments were performed with two different proteases, proteinase-K and trypsin. Being different from the other proteinases, proteinase K presents keratin hydrolyzing activity. Cleaving peptide bonds of aliphatic, aromatic, and hydrophobic residues, proteinase K reveals to be a broad spectrum (non-specific) proteinase [35]. On the other side, trypsin is a protease with high cleavage specificity, cleaving C-terminal to arginine or lysine residues.

Treatment with proteinase K (4.90 ± 0.14 log CFU/mL) and trypsin (4.96 ± 0.16 log CFU/mL) did not exert significant effects in *L*. *rhamnosus* (5.06 ± 0.03 log CFU/mL) adhesion to HaCaT cells (Figure 9), suggesting that this bacterium does not utilize an adhesin to attach to keratinocytes. No significant statistical differences between the pre-treatment groups compared to the control group were found (*p* > 0.05). The results for *L*. *rhamnosus* and *S*. *aureus* together revealed that the probiotic bacteria was not affected by the protease pre-treatment, as well.

On the other hand, the pathogenic *S*. *aureus* demonstrated a slight decrease with the pre-treatment by both proteases, proteinase K (5.49 ± 0.07 log CFU/mL) and trypsin (5.34 ± 0.40 log CFU/mL), when compared to the control (6.04 ± 0.02 log CFU/mL) with a significative difference with respect to the control group (*p* < 0.05). This indicates that *S. aureus* may utilize a cell surface protein to adhere to HaCaT cells. Equally, *S*. *aureus* with *L*. *rhamnosus* (in BHI medium), was unable to attach to keratinocytes as well as the control, meaning that the competition for the adhesion sites decreased its adhesion when compared to the results for *S*. *aureus* alone (*p* < 0.05).

### 3.6. Human Skin Models

To evaluate the influence of the probiotic *L*. *rhamnosus* in wound skin infections by the pathogen *S*. *aureus*, an ex vivo trial using human skin equivalents was performed and monitored for 11 days, as explained previously. Skin wounds are extremely susceptible to bacterial infection, which may impede the wound healing process and lead to systemic complications [36]. Figure 10 represents the bacterial values obtained in two distinct analyses: first, the results for the monitorization collected through swabbing (Figure 10A), and next the total counts after destruction of the skin models (Figure 10B).

It was possible to observe that in both situations, swab analysis and sacrificing the models, the probiotic *L*. *rhamnosus* was present in lower numbers than *S*. *aureus*. Concerning the pathogen, *S*. *aureus* tended to increase during this time, as described in a previous study utilizing tissue-engineered models [37]. During bacterial growth, some fluctuations in the CFU/mL occurred (the most significant is on the first wound in MRS medium third sampling, with a decrease of 1.03 log followed by an increase of 1.45 log); this fact can be due to sample collection through swabbing since the amount of collected bacteria is not the same. Nevertheless, the hypothesis that instigated this assay reveals promising, since the model with the wounds infected with the pathogen and later inoculated with the probiotic showed lower numbers of *S*. *aureus* in both analyses when compared to the wound with no probiotic treatment.

Besides the bacterial counting, wound healing was monitored through macroscopic observation and photographs were taken every day, several of them are present in Figure 11. Through macroscopic monitorization of the skin models, the control skin model, with no bacterial addition, had higher healing of the wound than the models with bacterial addition, proving that bacteria could delay the wound healing process in human skin models, in line with what is observed in the skin [36]. In injuries involving only the epidermis, the regeneration of the deep tissue is rapid with re-epithelialization occurring in 7 to 11 days [38]. Although in the assay it was not possible to see the regeneration of the surface on the control sample, corresponding to the *stratum corneum*, the re-epithelialization of the underlying layer was notorious.

The model with the infected wound and treated with *L*. *rhamnosus* revealed an advanced healing process. Moreover, this wound presented a slightly orange color (after day 4), a fact that can be related to acidification of the medium, since *L*. *rhamnosus* produces lactic acid. *L*. *rhamnosus* did not delay the wound healing process, corroborating with a study in which *L*. *rhamnosus* lysate was applied to reconstructed human keratin, demonstrating an improvement in skin barrier function by increasing the expression of tight junction proteins [39].

Although both wounds in the skin model infected with *S*. *aureus* showed no healing progress during the experiment, a visible amount of exudate was present and was formed as a white halo around the wounds, turning bigger over the days. The delay in the wound healing process caused by this pathogen is well described and the possible causes behind that have been studied using mice models [40].

Finally, the skin model with infected wounds by both bacteria revealed a wound healing progress but was more discreet when compared to the control or the model inoculated with *L*. *rhamnosus* alone. Similarly, in the co-infected model, a white halo around the wounds was also formed, although less evident. The exudate produced was in a diminutive amount when compared to the model infected with the pathogen alone. The macroscopic analysis was in agreement with the bacterial cell count analysis, indicating that the probiotic *L*. *rhamnosus* could be used as an adjuvant in *S*. *aureus* wound infections.

## 4. Conclusions

The tested probiotics proved capable of keratinocyte invasion. Adhesion to keratinocytes is greatly dependent on the order of bacterial addition. *L*. *rhamnosus* and *P*. *innocua* were the most effective in inhibiting pathogen adhesion to keratinocytes, reinforcing that the probiotics under study could be used to reduce the attachment of these pathogens to skin keratinocytes in the referred situations, constituting an important finding, since cell adhesion is considered to be the first step in bacterial colonization, leading to infection. However, pathogens can decrease probiotic adhesion too, probably owing to competition for the binding sites. Probiotics use carbohydrates to mediate the adhesion while *S*. *aureus* utilizes proteins (adhesins).

Ex vivo assays showed significant differences between models infected only with the pathogen and models infected with the pathogen and the probiotic *L*. *rhamnosus* as a post-treatment after 24 h. Visual improvement of the wound treated with *L*. *rhamnosus* addition was observable, particularly in wound size and exudate production, besides decreasing the number of viable *S*. *aureus*.

The findings described in this research show that the studied probiotics could be used topically with relevant pathogen inhibition ability, representing an important adjuvant in the clinical approach to treating patients with skin dysbiosis such as atopic dermatitis, acne, or conditions leading to cutaneous infections, as complicated wounds.

## Figures and Tables

**Figure 1 biology-11-01372-f001:**
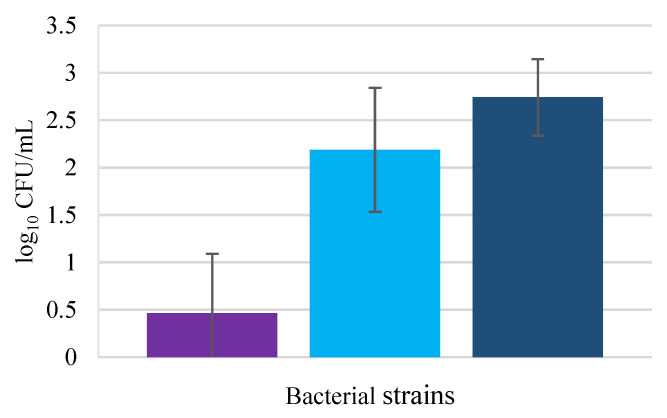
Pathogenic bacteria invasion of HaCat cells; *Escherichia coli* (■), *Pseudomonas aeruginosa* (■), and *Staphylococcus aureus* (■). Error bars are ± standard deviation.

**Figure 2 biology-11-01372-f002:**
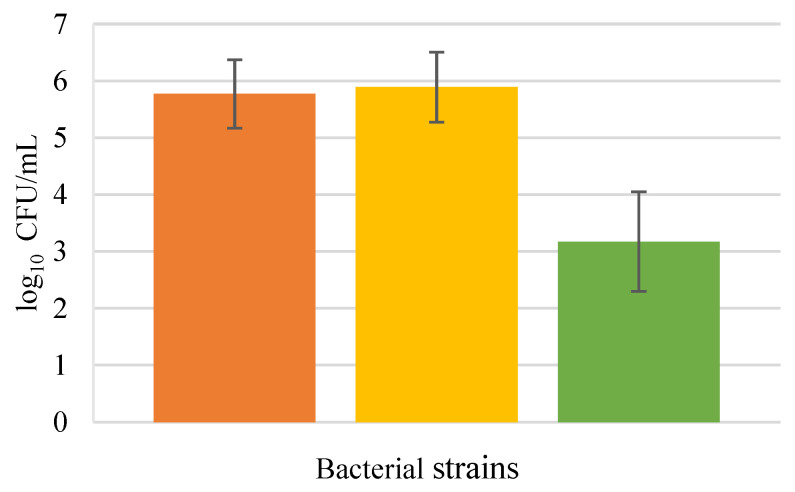
Probiotic bacteria invasion of HaCat cells in *Lacticaseibacillus rhamnosus* (■), *Lactobacillus delbrueckii* subsp. *bulgaricus* (■), and *Propioniferax innocua* (■). Error bars are ± standard deviation.

**Figure 3 biology-11-01372-f003:**
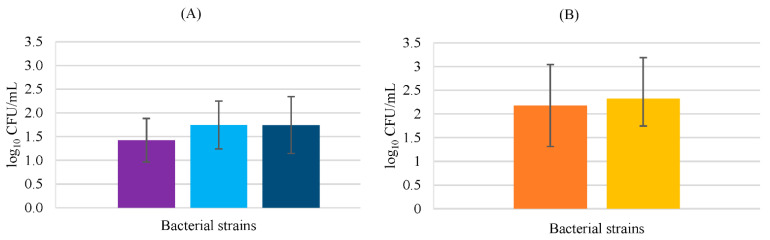
HaCaT cell invasion by *P*. *aeruginosa* in the presence of probiotic bacteria. *P*. *aeruginosa* counts in BHI medium (**A**) *Lacticaseibacillus rhamnosus* (■), *Lactobacillus delbrueckii* subsp. *bulgaricus* (■), and *Propioniferax innocua* (■); counts in MRS medium (**B**) *Lacticaseibacillus rhamnosus* (■), *Lactobacillus delbrueckii* subsp *bulgaricus* (■). Error bars are ± standard deviation.

**Figure 4 biology-11-01372-f004:**
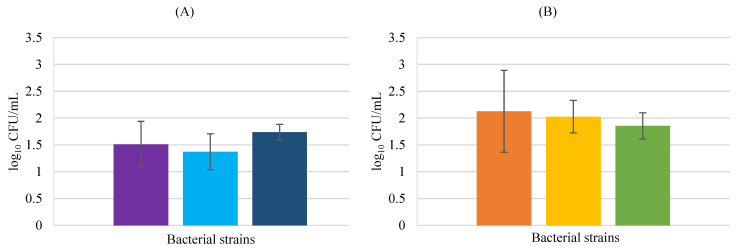
HaCaT cell invasion by *S. aureus* in the presence of probiotic bacteria. *S. aureus* counts in BHI medium (**A**) *Lacticaseibacillus rhamnosus* (■), *Lactobacillus delbrueckii* subsp. *bulgaricus* (■), and *Propioniferax innocua* (■); counts in MRS medium (**B**) *Lacticaseibacillus rhamnosus* (■), *Lactobacillus delbrueckii* subsp. *bulgaricus* (■), and *Propioniferax innocua* (■). Error bars are ± standard deviation.

**Figure 5 biology-11-01372-f005:**
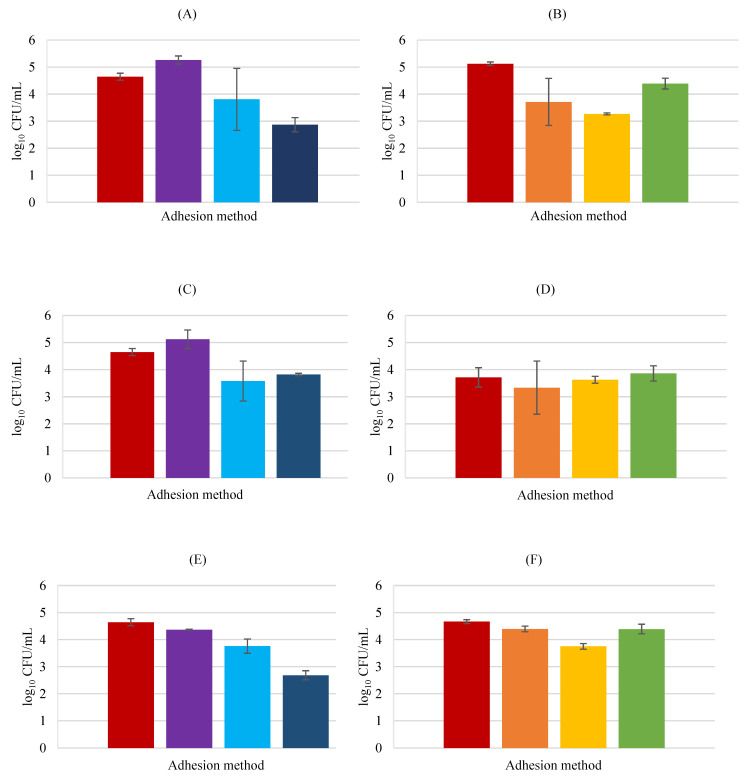
HaCaT cell adhesion by *E*. *coli* in the presence of probiotics. *E*. *coli* plus *L*. *rhamnosus* (**A**,**B**) *E*. *coli* plus *L*. *delbrueckii* (**C**,**D**) *E*. *coli* plus *P. innocua* (**E**,**F**). In BHI medium counts (**A**,**C**,**E**) exclusion (■), displacement (■), and competition (■); MRS medium counts (**B**,**D**,**F**) exclusion (■), displacement (■), and competition (■); and respective control (■). Error bars are ± standard deviation.

**Figure 6 biology-11-01372-f006:**
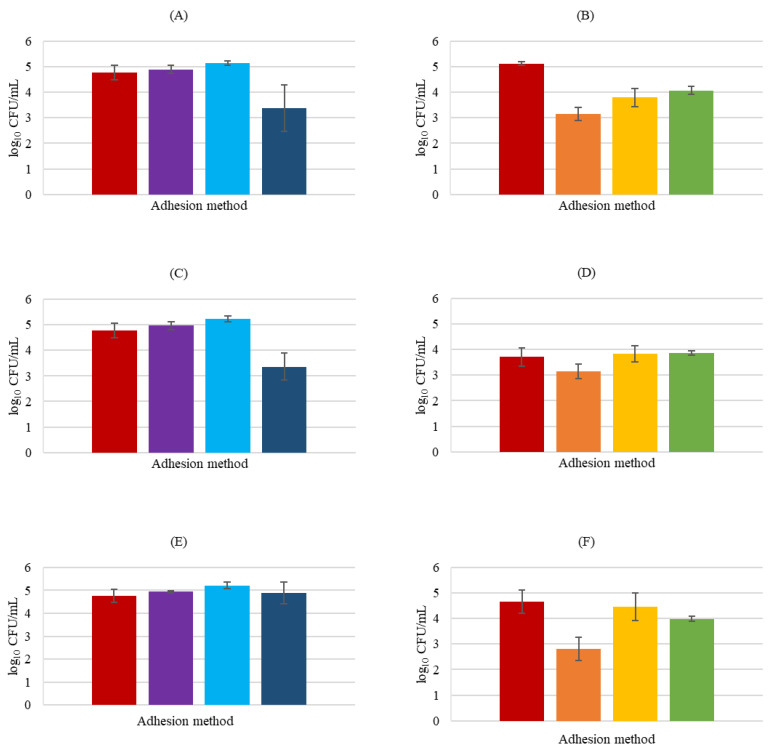
HaCaT cell adhesion by *P*. *aeruginosa* in the presence of probiotics. *P*. *aeruginosa* plus *L*. *rhamnosus* (**A**,**B**), *P*. *aeruginosa* plus *L*. *delbrueckii* (**C**,**D**), and *P*. *aeruginosa* plus *P. innocua* (**E**,**F**). In BHI medium counts (**A**,**C**,**E**) exclusion (■), displacement (■), and competition (■); MRS medium counts (**B**,**D**,**F**) exclusion (■), displacement (■), and competition (■); and respective bacterial control (■). Error bars are ± standard deviation.

**Figure 7 biology-11-01372-f007:**
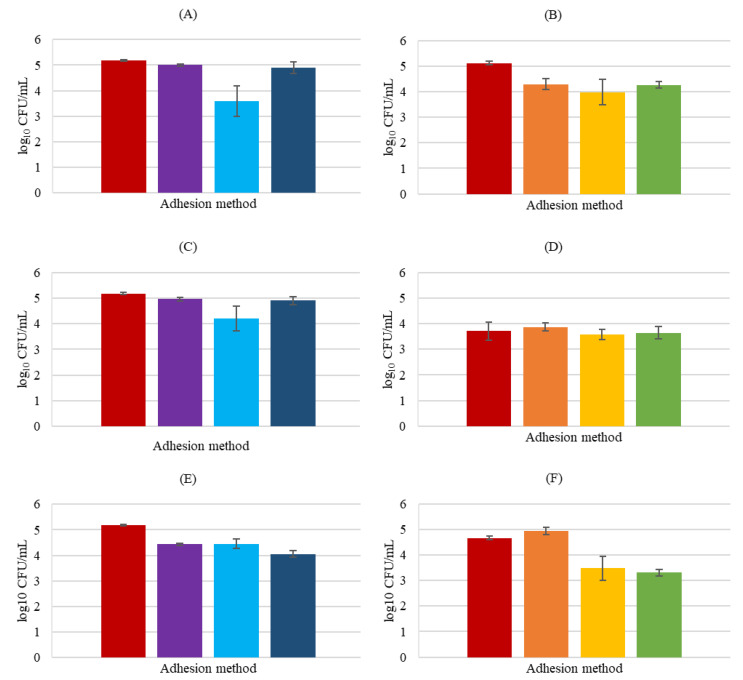
HaCaT cell adhesion by *S*. *aureus* in the presence of probiotics. *S*. *aureus* plus *L*. *rhamnosus (***A**,**B**), *S*. *aureus* plus *L*. *delbrueckii* (**C**,**D**), and *S*. *aureus* plus *P. innocua* (**E**,**F**). In BHI medium counts (**A**,**C**,**E**) exclusion (■), displacement (■), and competition (■); MRS medium counts (**B**,**D**,**F**) exclusion (■), displacement (■), and competition (■); and respective control (■). Error bars are ± standard deviation.

**Figure 8 biology-11-01372-f008:**
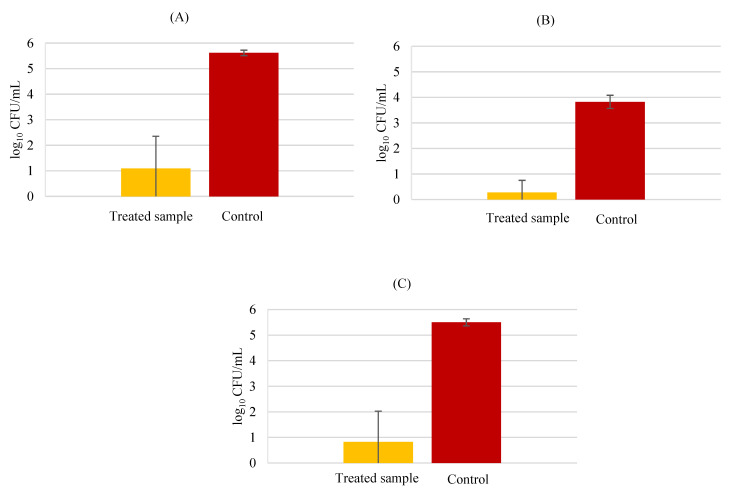
Carbohydrate analysis in adhesion to HaCaT. (**A**) *Lacticaseibacillus rhamnosus*; (**B**) *Lactobacillus delbrueckii* subsp. *Bulgaricus*; (**C**) and *Propioniferax innocua*. Error bars are ± standard deviation.

**Figure 9 biology-11-01372-f009:**
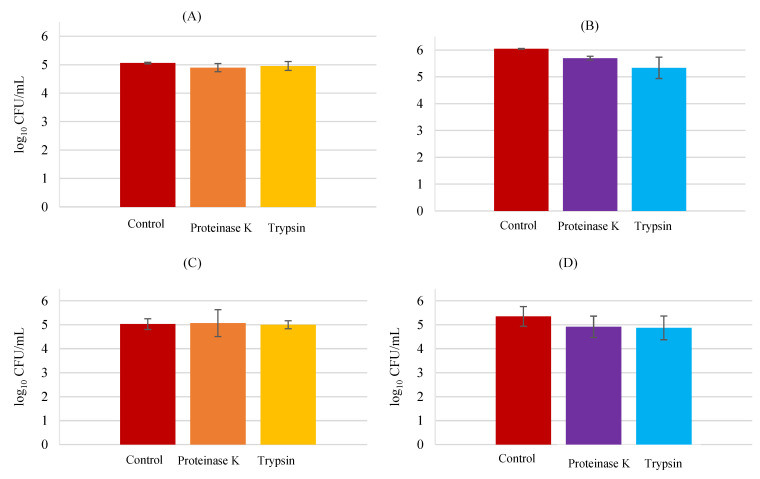
Protein analysis in adhesion to HaCaT; *L*. *rhamnosus* (**A**) *S*. *aureus* (**B**) *L*. *rhamnosus* plus *S*. *aureus*, MRS counts (**C**) *L*. *rhamnosus* plus *S*. *aureus*, BHI counts (**D**). Error bars are ± standard deviation.

**Figure 10 biology-11-01372-f010:**
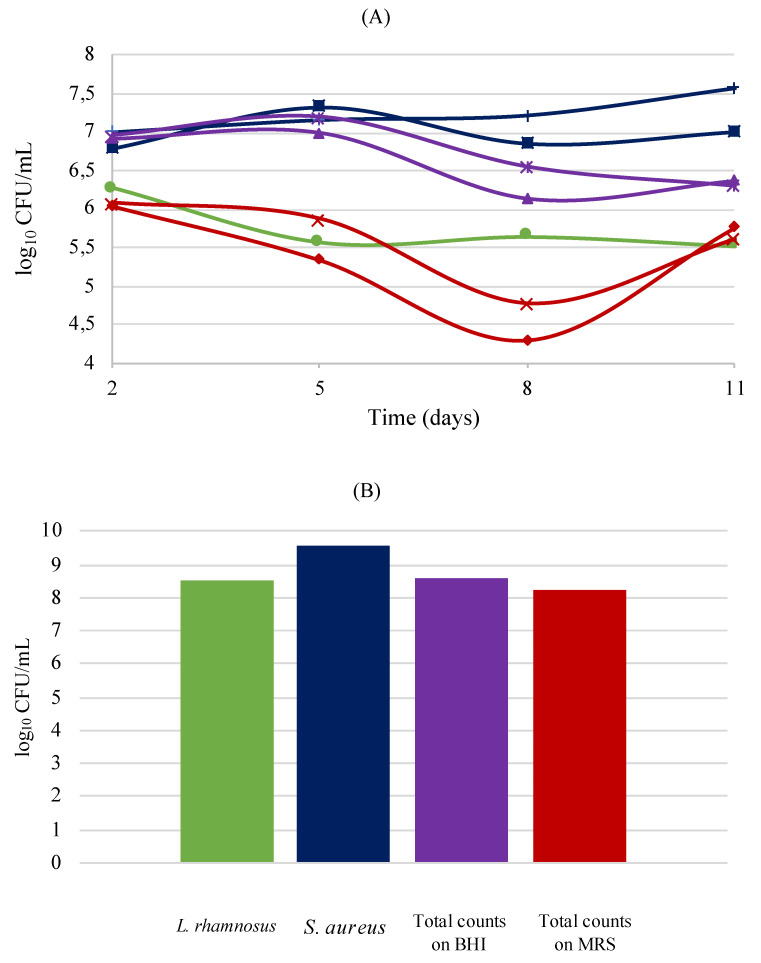
Bacterial counts in human skin equivalents. Counts on the skin models collected with swab (**A**): *S*. *aureus* first wound (+) *S*. *aureus* second wound (■) *S*. *aureus* plus *L*. *rhamnosus*—first wound (✴) *S*. *aureus* plus *L*. *rhamnosus*—second wound (▲) *S*. *aureus* plus *L*. *rhamnosus*—first wound (✕) *S*. *aureus* plus *L*. *rhamnosus*—second wound (♦) *L*. *rhamnosus* (●). Bacterial counts after sacrificing the models (**B**).

**Figure 11 biology-11-01372-f011:**
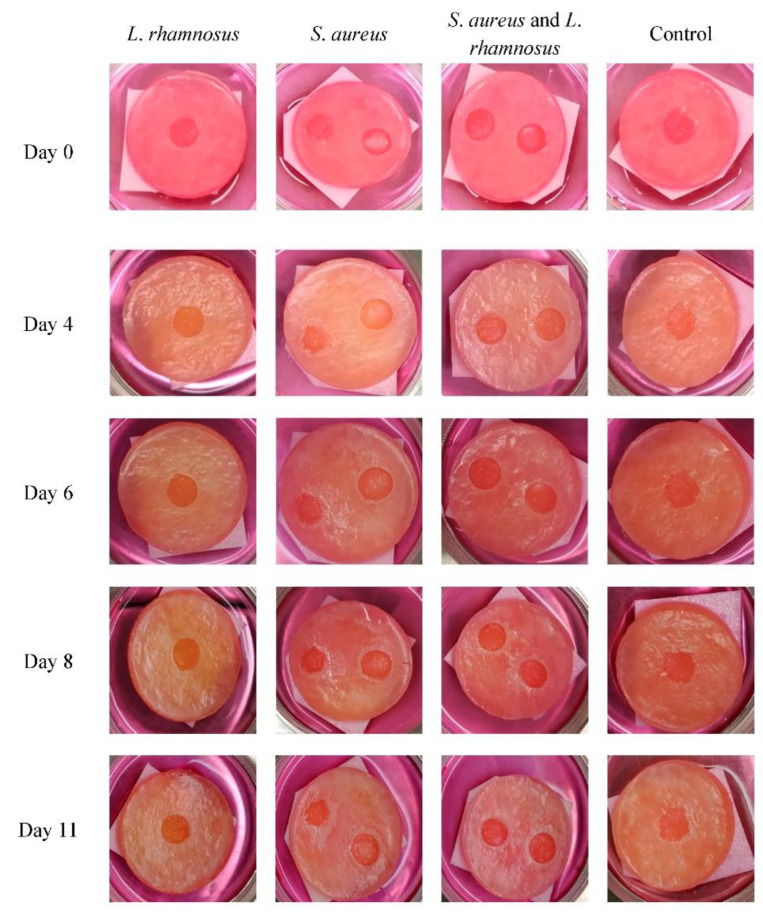
Progression of the wound healing process in skin models. Macroscopic photo images of the human skin equivalents from day 0 to day 11. After 4 days, an orange color is observable due to lactic acid formation in the infected models, reflecting a drop in the pH. After day 6, contraction of the wound size is visible only in the models infected with *L. rhamnosus*. Models infected with *S. aureus* showed the presence of exudate visible as a white precipitate, as well as a delayed healing.

## Data Availability

Data is contained within the article.

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
