# Peer review of "Probiotic Adhesion to Skin Keratinocytes and Underlying Mechanisms"

_biology, 2022, doi:10.3390/biology11091372_

Round 1
Reviewer 1 Report
The study aimed to investigate the ability of the probiotics Lactobacillus delbrueckii DSM 20081, Lactobacillus rhamnosus DSM 20021 and Propioniferax innocua DSM 8251 to inhibit adhesion and invasion of Escherichia coli, Pseudomonas aeruginosa and Staphylococcus aureus using human HaCaT keratinocytes. The authors observed the used strain of Lactobacillus rhamnosus with promising results. However, there is a need for some revisions to the manuscript. Below are some suggestions and questions:
Lines 50 – 52: Please, add a reference to the sentence;
Line 57: Please authors should use the term “Gram-positive” instead of “Gram-stain-positive”;
Line 80: Please delete “microbiome”, as the term dysbiosis already means an unbalanced microbiome;
- Add the correct symbol for the degrees Celsius throughout the methodology;
Methodology: I suggest that authors add the letters DSM before each number referring to the probiotics, as well as the strain codes of the pathogenic bacteria studied;
Lines 103 – 107: The names of microorganisms should be in italics. In addition, there was a change in the taxonomy of the Lactobacillus genus and many articles are already being published with the changes. Therefore, I suggest that the authors analyze whether it would not be appropriate to carry out this update in the designation of the species Lactobacillus delbrueckii and Lactobacillus rhmanosus (Zheng et al. Int J Syst Evol Microbiol 2020, 70, (4), 2782-2858, doi:10.1099 /ijsem.0.004107.);
Lines 139 – 154: In the cell invasion assay it is important to determine in advance whether the microorganisms to be tested are sensitive to gentamicin. This care is essential to avoid errors in the interpretation of results, if the assay was performed with a bacterial strain resistant to gentamicin. If so, this could explain the “higher invasion” (mentioned in lines 281-284 and demonstrated in the results) of the two Lactobacillus strains. The authors should clarify whether this has been done or whether the susceptibility profile of the strains has been previously determined by others.
Figure 11: it is difficult to differentiate by visual analysis from the advanced process of wound healing by probiotics in skin models;
Author Response
Comments and Suggestions for Authors
The study aimed to investigate the ability of the probiotics Lactobacillus delbrueckii DSM 20081, Lactobacillus rhamnosus DSM 20021 and Propioniferax innocua DSM 8251 to inhibit adhesion and invasion of Escherichia coli, Pseudomonas aeruginosa and Staphylococcus aureus using human HaCaT keratinocytes. The authors observed the used strain of Lactobacillus rhamnosus with promising results. However, there is a need for some revisions to the manuscript. Below are some suggestions and questions:
Lines 50 – 52: Please, add a reference to the sentence;
Reference added (Kaliyeva et al., 2022). This is now reference 7, all other numbering was changed accordingly.
Line 57: Please authors should use the term “Gram-positive” instead of “Gram-stain-positive”;
Agreed. Duly changed.
Line 80: Please delete “microbiome”, as the term dysbiosis already means an unbalanced microbiome;
Done.
Add the correct symbol for the degrees Celsius throughout the methodology;
Done. All corrected.
Methodology: I suggest that authors add the letters DSM before each number referring to the probiotics, as well as the strain codes of the pathogenic bacteria studied;
Done.
Lines 103 – 107: The names of microorganisms should be in italics. In addition, there was a change in the taxonomy of the Lactobacillus genus and many articles are already being published with the changes. Therefore, I suggest that the authors analyze whether it would not be appropriate to carry out this update in the designation of the species Lactobacillus delbrueckii and Lactobacillus rhmanosus (Zheng et al. Int J Syst Evol Microbiol 2020, 70, (4), 2782-2858, doi:10.1099 /ijsem.0.004107.);
In the original document, the names of the microorganisms were italicized. Must have disformatted with the conversion. Thank you. According to the new nomenclature, Lactobacillus delbrueckii subsp. bulgaricus remains the same while Lactobacillus rhamnosus has changed to Lacticaseibacillus rhamnosus. Changed throughout the manuscript.
Lines 139 – 154: In the cell invasion assay it is important to determine in advance whether the microorganisms to be tested are sensitive to gentamicin. This care is essential to avoid errors in the interpretation of results, if the assay was performed with a bacterial strain resistant to gentamicin. If so, this could explain the “higher invasion” (mentioned in lines 281-284 and demonstrated in the results) of the two Lactobacillus strains. The authors should clarify whether this has been done or whether the susceptibility profile of the strains has been previously determined by others.
The test for antimicrobial susceptibility to gentamicin was not conducted, however it has been reported elsewhere (Coppola et al., 2005) and shows that this particular DSM strain of Lactobacillus rhamnosus is susceptible to gentamicin, which proves that antibiotic resistance is not the cause for higher invasion. We consider this an important consideration and have therefore, clarified it in the text.
Reference added: Raffaele Coppola, Mariantonietta Succi, Patrizio Tremonte, Anna Reale, Giovanni Salzano, et al. Antibiotic susceptibility of Lactobacillus rhamnosus strains isolated from Parmigiano Reggiano cheese. Le Lait, INRA Editions, 2005, 85 (3), pp.193-204.
Regarding Lactobacillus delbrueckii ssp. bulgaricus we could not find in the literature if this particular strain from DSM was susceptible or resistant to gentamicin. However, for other strains, the reports found in the literature are inconsistent (see refs below, please), with results on antibiotic resistance depending on the origin of the isolates, methods applied in studies, and/or culture conditions such as media used for culturing. However, for the purpose intended in this work it is not relevant since this isolate showed lower invasion values.
Tavsanli H., Mus T.E., Cetinkaya F., Ayanoglu E., Cibik R. (2021): Isolation of Lactobacillus delbrueckii spp. bulgaricus and Streptococcus thermophilus from nature: Technological characterisation and antibiotic resistance. Czech J. Food Sci., 39: 305–311.
Shani N, Oberhaensli S, Berthoud H, Schmidt RS, Bachmann HP. Antimicrobial Susceptibility of Lactobacillus delbrueckii subsp. lactis from Milk Products and Other Habitats. Foods. 2021 Dec 18;10(12):3145. doi: 10.3390/foods10123145. PMID: 34945696; PMCID: PMC8701367.
Figure 11: it is difficult to differentiate by visual analysis from the advanced process of wound healing by probiotics in skin models;
We agree as it is only visual/morphological information, and have therefore included more information in the caption to guide the reader.
Reviewer 2 Report
· Authors should apply new nomenclature for Lactobacillus species, as it was divided into 25 genera. See: LACTOTAX webpage http://lactotax.embl.de/wuyts/lactotax/ and
Zheng J, Wittouck S, Salvetti E, et al. A taxonomic note on the genus Lactobacillus: Description of 23 novel genera, emended description of the genus Lactobacillus Beijerinck 1901, and union of Lactobacillaceae and Leuconostocaceae. Int J Syst Evol Microbiol. 2020. doi: 10.1099/ijsem.0.004107.
· The Authors should consider whether the tested strains are actually probiotic? If not, do not use this term, but e.g. probiotic candidates.
· The Introduction is written as if it were divided into unrelated sections. Please link the various topics discussed in the introduction in a logical way.
· Emphasise the novelty of the study.
· Section 2.1. - give strain numbers/symbols for pathogens.
· What was the viability of HaCaT cells taken to the experiments?
· Line 154: how they were counted? Name the method.
· Section 2.5. I do not understand it. Justify/explain please conducting the experiment.
· Do I think that all figures are missing the X and Y axes and captions to the Y axis? Therefore the graphs are hardly legible.
· English grammar and style must be improved.
· Figure 11 caption is of poor quality. Otherwise it is not informative. Give more details.
Author Response
- Authors should apply new nomenclature for Lactobacillusspecies, as it was divided into 25 genera. See: LACTOTAX webpage http://lactotax.embl.de/wuyts/lactotax/ and
Zheng J, Wittouck S, Salvetti E, et al. A taxonomic note on the genus Lactobacillus: Description of 23 novel genera, emended description of the genus Lactobacillus Beijerinck 1901, and union of Lactobacillaceae and Leuconostocaceae. Int J Syst Evol Microbiol. 2020. doi: 10.1099/ijsem.0.004107.
Thank you. This was corrected.
- The Authors should consider whether the tested strains are actually probiotic? If not, do not use this term, but e.g. probiotic candidates.
Most Lactobacilli have all the attributes of standard probiotic bacteria. In fact, there are studies with these particular strains showing that these are in fact probiotic.
Mater DD, Bretigny L, Firmesse O, Flores MJ, Mogenet A, Bresson JL, Corthier G. Streptococcus thermophilus and Lactobacillus delbrueckii subsp. bulgaricus survive gastrointestinal transit of healthy volunteers consuming yogurt. FEMS Microbiol Lett. 2005 Sep 15;250(2):185-7. doi: 10.1016/j.femsle.2005.07.006. PMID: 16099606.
Abedi D, Feizizadeh S, Akbari V, Jafarian-Dehkordi A. In vitro anti-bacterial and anti-adherence effects of Lactobacillus delbrueckii subsp bulgaricus on Escherichia coli. Res Pharm Sci. 2013 Oct;8(4):260-8. PMID: 24082895; PMCID: PMC3757591.
Succi, M. et al. 2017. Pre-cultivation with Selected Prebiotics Enhances the Survival and the Stress Response of Lactobacillus rhamnosus Strains in Simulated Gastrointestinal Transit. Front. Microbiol., https://doi.org/10.3389/fmicb.2017.01067
- The Introduction is written as if it were divided into unrelated sections. Please link the various topics discussed in the introduction in a logical way.
Done. Thank you for the suggestion. We have re-written some parts to facilitate understanding. We hope it is now made clear.
- Emphasise the novelty of the study.
Also done in the end of the introduction.
- Section 2.1. - give strain numbers/symbols for pathogens.
These strains are not numbered as they are part of our internal collection of isolates.
- What was the viability of HaCaT cells taken to the experiments?
2 x 105 (in M&M section, line 328)
- Line 154: how they were counted? Name the method.
Trypan blue exclusion method. Added to text.
- Section 2.5. I do not understand it. Justify/explain please conducting the experiment.
In order to investigate the possibility of a cell-surface carbohydrate being implicated in the mechanism of the studied probiotic strains adhesion to HaCaT cells, Sodium-meta periodate was used. It is a chemical compound that oxidizes cell surface carbohydrates, being considered a useful tool to explore the structures of carbohydrates and compounds containing oxidizable functions (Hudson and Barker, 1967; Prince, 2012). It has been used in studies to investigate probiotic adhesion to cells (Prince, 2012). Afterwards, utilizing the plate count method only the viable adherent bacteria were able to be counted. Knowing that a more reduced number of adherent bacteria in the pre-treated group compared to control indicates that this compound may inhibit (through oxidation) the carbohydrates that could be behind the probiotic adhesion to keratinocyte cells.
- Do I think that all figures are missing the X and Y axes and captions to the Y axis? Therefore the graphs are hardly legible.
The graphs were all re-done to include both axis.
- English grammar and style must be improved.
Thank you. We have duly changed the text so it is now hopefully clearer to the reader.
- Figure 11 caption is of poor quality. Otherwise it is not informative. Give more details.
The caption was re-written to give more detailed information. Hope it is now suitable for better comprehension.
Round 2
Reviewer 1 Report
The authors accepted the suggestions and clarified the doubts. The manuscript has improved and can now be accepted for publication.
Author Response
Thank you for your valuable review.
Reviewer 2 Report
I would give the title to X axis "Strains" in the graphs.
Author Response
We have accepter this suggestion and added labels to the xx axis in all Figures. We thank you for this suggestion, since it is in fact easier to understand.